# In Vitro, In Vivo, and In Silico Analyses of Molecular Anti-Pigmentation Mechanisms of Selected Thai Rejuvenating Remedy and Bioactive Metabolites

**DOI:** 10.3390/molecules28030958

**Published:** 2023-01-18

**Authors:** Sukanya Dej-adisai, Nitinant Koyphokaisawan, Chatchai Wattanapiromsakul, Wanlapa Nuankaew, Tong Ho Kang, Thanet Pitakbut

**Affiliations:** 1Department of Pharmacognosy and Pharmaceutical Botany, Faculty of Pharmaceutical Sciences, Prince of Songkla University, Hat Yai 90112, Songkhla, Thailand; 2Department of Oriental Medicinal Biotechnology, Graduate School of Biotechnology, College of Life Sciences, Kyung Hee University, Yongin-si 17104, Gyeonggi-do, Republic of Korea; 3Pharmaceutical Biology, Department of Biology, Friedrich-Alexander-Universität Erlangen-Nürnberg (FAU), 91058 Erlangen, Germany

**Keywords:** molecular anti-pigmentation mechanisms, tyrosinase inhibition, MC1R inhibition, Thai rejuvenating remedy, glabridin, ethyl *p*-methoxycinnamate, molecular docking, plant-derived tyrosinase inhibitor

## Abstract

Thai rejuvenating remedies are mixed herbal formulas promoting longevity. Due to the complexity, the biological activities of these remedies are minimal. Therefore, in this study, the authors evaluated the anti-pigmentation effect at the molecular level of the selected Thai rejuvenating remedy to fulfill the knowledge gap. First, the authors found that the selected remedy showed promising activity against the tyrosinase enzyme with an IC_50_ value of 9.41 µg/mL. In the comparison, kojic acid (positive control) exhibited an IC_50_ value of 3.92 µg/mL against the same enzyme. Later, the authors identified glabridin as a bioactive molecule against tyrosinase with an IC_50_ value of 0.08 µg/mL. However, ethyl *p*-methoxycinnamate was the most abundant metabolite found in the remedy. The authors also found that the selected remedy and glabridin reduced the melanin content in the cell-based assay (B16F1) but not in the zebrafish larvae experiment. Finally, the authors conducted a computational investigation through molecular docking proposing a theoretical molecular interplay between glabridin, ethyl *p*-methoxycinnamate, and target proteins (tyrosinase and melanocortin-1 receptor, MC1R). Hence, in this study, the authors reported the molecular anti-pigmentation mechanism of the selected Thai rejuvenating remedy for the first time by combining the results from in silico, in vitro, and in vivo experiments.

## 1. Introduction

Thai rejuvenating remedy, also known as “Yaa-Aa-Yu-Wat-Ta-Na” in Thai, has been claimed to slow down the aging process and to nourish, reboot, and restore balance to the human body [1,2]. Typically, each Thai rejuvenating remedy consists of a specific medicinal plant mixture in various proportions, and each unique remedy claims specific therapeutic properties. An idea of a complex herbal mixture is based on a holistic medical concept used by traditional Thai medicine practitioners [3]. However, this concept is challenging when applying a modern scientific approach, such as reductionism [4]. Therefore, only a few remedies have been investigated scientifically; thus, the resulting reliable databases used for supporting the utilization of Thai rejuvenating remedies correctly and safely are limited. Nevertheless, it is vital to search for pharmaceutical products to prevent, restore, and maintain human body function for the elderly.

Aging has a pronounced effect on the deceleration of the functions and structures of multiple human systems and organs [5]. For example, the skin causes various structural components, such as the thinning of the epidermis and dermis, fragmentation of collagen and elastic fibers, and reduction in cell healing and cell restoration [6,7]. These changes lead to skin problems, for instance, one of the most common problems is solar lentigo (a hyperpigmented spot on the skin) [8,9]. Research studies have found that aging and prolonged UV exposure are common risk factors [8,10]. These factors cause pigment overproduction, known as hyperpigmentation. Hyperpigmentation disorder can lead to other severe medical conditions, such as skin cancer [11], and it also causes psychological problems, such as emotional distress [12]. Therefore, research studies have suggested two main approaches to prevent or reduce melanin biosynthesis [13,14]. The first is to avoid risk factors that activate melanin (pigment) production by applying skin care products with UV protection. The second is to use tyrosinase inhibitors to block melanin biosynthesis [13,14]. Tyrosinase is an essential enzyme in melanin biosynthesis. Therefore, using a tyrosinase inhibitor can prevent hyperpigmentation [15,16].

The recent literature reviews have shown that topical therapy, such as creams, is the first line of treatment for melanin overproduction [13,14]. Interestingly, plant-derived tyrosinase inhibitors, such as kojic acid and arbutin, are commonly used in a prescribed topical anti-pigmentation cream [14]. This information hints at plants as a potential source for tyrosinase inhibitor discovery [17]. However, using randomly selected medicinal plants to evaluate anti-pigmentation effects does not seem rational. Therefore, the authors decided to screen the anti-pigmentation activities from known Thai rejuvenating remedies since it has been claimed that they have anti-aging and skin restoring effects. Additionally, Thai people have used these remedies for centuries [1], suggesting a minimal toxicity effect. Therefore, it serves as an advantage point for cell-based and in vivo studies.

In this study, sixty-two Thai rejuvenating remedies were screened for anti-tyrosinase activity. Later, the remedy with the highest tyrosinase inhibitory activity was selected and subjected to a chemical analysis to determine the bioactive and most abundant metabolite presented in the remedy. After that, the authors further examined the potential of the selected remedy in cell-based and zebrafish larvae anti-pigmentation assays. Finally, bio- and chemoinformatic experiments, such as multiple alignment, phylogenetic analysis, 3D structural alignment, and molecular docking, were performed to provide theoretical support. The results from this study provide an intensive scientific background of molecular anti-pigmentation mechanisms of the selected Thai rejuvenating remedy and will be of benefit to further studies in the future.

## 2. Results

### 2.1. Anti-Mushroom Tyrosinase Activity of the Selected Remedy

In this experiment, the authors screened the anti-mushroom tyrosinase activity of sixty-two Thai rejuvenating remedies and provided the result of this screening step in a Appendix A. The authors aimed to select the remedy exhibiting the most potent activity against mushroom tyrosinase. As a result, remedy No. 11 was the most promising one. Therefore, the authors selected remedy No. 11 to investigate the anti-tyrosinase activity further.

The authors moved forward to evaluate the anti-mushroom tyrosinase activity of series solvent extracts from remedy No. 11 and compared them to crude extract (80% ethanol). In this experiment, the authors used a 20 µg/mL concentration as the screening concentration. As presented in Table 1, ethyl acetate extract showed the same inhibitory potency as a crude extract. Ethyl acetate extract exhibited 71.06 ± 1.38% inhibition at a screening concentration, with an IC_50_ value (a concentration of inhibitor inhibiting half of the enzyme activity) of 9.98 µg/mL. At the same time, a crude extract showed 70.44 ± 0.23% inhibition at the same concentration with the same range of an IC_50_ value of 9.41 µg/mL.

Later, the authors investigated the anti-mushroom tyrosinase of remedy No. 11 ingredients (Table 2). Interestingly, 80% ethanol extract of *Glycyrrhiza glabra* was the only ingredient inhibiting tyrosinase activity with 80.07 ± 1.04% inhibition at 20 µg/mL concentration and an IC_50_ value of 3.74 µg/mL. The activity from *G. glabra* extract was similar to kojic acid, IUPAC name 5-hydroxy-2-(hydroxymethyl)pyran-4-one (one of the external standards), showing 85.93 ± 0.24% inhibition at the same concentration and an IC_50_ value of 3.92 µg/mL. However, *G. glabra* extract still exhibited a much weaker activity than *Artocarpus lacucha* wood water extract (95.02 ± 0.36% inhibition at a screening concentration with an IC_50_ value of 0.10 µg/mL). Finally, the author evaluated the anti-mushroom tyrosinase of glabridin, IUPAC name 4-[(3R)-8,8-dimethyl-3,4-dihydro-2H-pyrano [2,3-f]chromen-3-yl]benzene-1,3-diol, (a primary metabolite derived from *G. glabra*) and found that an IC_50_ value of glabridin was 0.08 µg/mL. It was slightly more potent than the IC_50_ value of *A. lacucha* wood water extract, our most potent external positive control.

### 2.2. Chemical Analysis and Metabolites Determination

#### 2.2.1. Gas Chromatography Coupled with Mass Spectroscopy (GC–MS)

Following the anti-tyrosinase activity, the authors analyzed chemical components presented in both remedies’ bioactive fractions (80% ethanolic and ethyl acetate extracts), Table 1, using GC–MS. The authors were only interested in the metabolites that passed the selection criteria. The first criterion was that the metabolites show more than 1% of the total components presented in the extract. The second criterion was that the percent match factor of analyzed metabolites exhibit higher than 70% prediction from the MS database, ensuring reliable metabolite determination.

As presented in Figure 1, the authors found that only two metabolites, ethyl-*p*-methoxycinnamate (IUPAC name: ethyl (*E*)-3-(4-methoxyphenyl)prop-2-enoate) and glabridin, were presented in both active fractions, inhibiting tyrosinase activity. However, since ethyl-*p*-methoxycinnamate presented with 22% and 20% in 80% ethanol and ethyl acetate extracts, respectively, the authors designed ethyl-*p*-methoxycinnamate as the major metabolite. In comparison, grabridin was only 2% in both extracts. Therefore, the authors designed grabridin as a minor metabolite. Finally, the authors proved the chemical structures of two metabolites found in bioactive fractions and kojic acid, one of the positive controls used in an enzyme-binding assay in Figure 2.

In the chemical analysis here, with earlier anti-tyrosinase activity (Table 2), glabridin was an active metabolite showing an intense anti-mushroom tyrosinase activity with an IC_50_ value of 0.08 µg/mL. Therefore, the authors decided to confirm only the presence of glabridin in 80% ethanol and ethyl acetate extracts using high-performance liquid column chromatography (HPLC) compared to a standard reference as further analysis.

#### 2.2.2. HPLC Analysis

The authors performed an HPLC analysis to confirm the presence of glabridin in both 80% ethanolic and ethyl acetate extract found in the GC–MS analysis earlier. Figure 3 showed that both extracts exhibited a similar peak at a retention time of 22 min under the photodiode array (PDA) detection at 230 nm to the glabridin standard peak. Therefore, based on the HPLC analysis, the authors confirmed the presence of glabridin in both 80% ethanolic and ethyl acetate extract from the selected remedy, which was in agreement with the previous GC–MS analysis.

### 2.3. Anti-Murine Melanoma (B16F1) Intracellular Tyrosinase Activity and Melanin Content

#### 2.3.1. Cytotoxicity Evaluation

Before performing cell-based assays, the authors evaluated the cytotoxicity effect of all test samples, ensuring that the results obtained in the following experiments did not interfere with this effect. In this evaluation, the authors determined a concentration of all test samples providing a percent cell viability of more than 80%, assuming minimal cytotoxicity effect. As a result, most samples showed a minimal cytotoxicity effect at a concentration of 25 µg/mL. However, only three samples, including *A. lacucha* wood water extract, *G. glabra* extract, and glabridin, exhibited a minimal cytotoxic effect at a concentration of 5 µg/mL (data did not show here). Therefore, as mentioned above, the authors used these concentrations for the following experiments.

#### 2.3.2. Anti-Murine Melanoma (B16F1) Intracellular Tyrosinase Activity

After an in vitro enzyme binding assay reported earlier, the authors evaluated the potency of the selected remedy extracts and active ingredients, including glabridin, in the cellular anti-tyrosinase model using murine melanoma (B16F1) cells. Here, the authors supplemented a cell culture medium with a hormone, upregulating tyrosinase enzyme expression. α-melanocyte-stimulating hormone, or α-MSH, was the hormone supplied in the medium at a concentration of 10 nM. According to the cytotoxicity report, the authors applied two concentrations (either 5 or 25 µg/mL) of samples in this experiment. After cultivation, the authors disrupted the cells and evaluated intercellular tyrosinase activity from a supernatant.

As a result, in Figure 4, only arbutin and *A. lacucha* extract (5 µg/mL concentration) slightly suppressed tyrosinase catalysis, with between 80% and 85% remaining activity (20% to 15% inhibition). Later, the authors further evaluated an IC_50_ value of arbutin, confirming a tyrosinase inhibitory activity of arbutin (positive control). The reason that the authors selected arbutin over *A. lacucha* extract was due to its more potent activity. Finally, the authors determined an IC_50_ value of arbutin of 119.10 µg/mL. However, the IC_50_ value of arbutin here exceeded the earlier cytotoxicity report. Therefore, the authors performed an additional cell toxicity assay of arbutin covering an IC_50_ concentration, and there was minimal toxicity up to a concentration of 200 µg/mL (Appendix A). On the other hand, kojic acid did not show any inhibitory activity.

Nearly all selected remedy extracts showed a slight to moderate inhibition against intercellular tyrosinase activity. The water extract of the selected remedy exhibited the weakest potency, with 94% remaining tyrosinase activity (6% inhibition), and the ethyl acetate was the most potent inhibition, with roughly 65% remaining enzyme function (35% inhibition). Only the hexane extract from the remedy did not exhibit any tyrosinase activity. In contrast, it promoted tyrosinase catalysis. Finally, the authors evaluated the remedy’s tyrosinase inhibitory activity from *G. glabra* extract and glabridin as active ingredients (at a concentration of 5 µg/mL). The authors found that *G. glabra* extract showed moderate activity, with approximately 75% remaining tyrosinase activity (25% inhibition). However, glabridin exhibited the most promising anti-tyrosinase activity above all samples, with only approximately 28% remaining enzyme function (72% inhibition). Later, the authors determined an IC_50_ value of glabridin against murine melanoma (B16F1) intracellular tyrosinase, and the glabridin IC_50_ value was 0.69 µg/mL. This IC_50_ value from glabridin was much stronger than the IC_50_ value from arbutin (positive control)—approximately 172 times stronger.

#### 2.3.3. Murine Melanoma (B16F1) Intracellular Melanin Content

Here, the authors compared melanin content after disrupting the murine melanoma (B16F1) cells among samples with control, supplemented with 10 nM α-MSH. Again, the authors used the same cytotoxicity report as earlier to determine each sample concentration used in this assay. Additionally, the authors compared the melanin content between low and high concentrations of α-MSH, 0.25 nM, and 10 nM, supplemented in the cultivation medium. This comparison was to ensure the α-MSH effect to melanogenesis stimulation, and the authors provided this result in a Appendix A.

Unlike a previous intracellular tyrosinase experiment, all positive controls suppressed melanin production here (Figure 4). Arbutin (5 µg/mL concentration) exhibited the most potent melanin reduction among these positive controls, with the lowest melanin content of 18.67 ± 1.26 µg/mL (46% reduction compared to the melanin content of the control). Kojic was the second-strongest inhibitor, containing 27.27 ± 0.68 µg/mL of melanin (21% depletion). Finally, *A. lacucha* extract was the less potent inhibitor, producing 30.29 ± 0.94 µg/mL of melanin (almost 13% deminution).

All remedy extracts showed a reduction in melanin content (Figure 5), starting from a slight to moderate reduction effect (4% to 46%). Remedy ethanolic extract exhibited a minimal reduction effect, with 33.26 ± 1.18 µg/mL melanin content (4% demotion). In contrast, ethyl acetate extract exhibited more of a reduction effect, with 18.49 ± 1.10 µg/mL melanin content (46% reduction). Later, according to the cytotoxic report, the authors further evaluated a melanin suppression effect from the active ingredients of the selected remedy—both *G. glabra* extract and glabridin with a concentration of 5 µg/mL. *G. glabra* extract slightly suppressed melanin production, with 28.57 ± 0.66 µg/mL melanin content (approximately 18% depletion). Again, glabridin was the most potent inhibitor above all samples, including the positive control, with a minimal melanin content of 9.56 ± 0.66 µg/mL (72% suppression). The authors provided all detailed data in a Appendix A.

#### 2.3.4. Correlation between Anti-Murine Melanoma (B16F1) Intracellular Tyrosinase Activity and Melanin Content

The authors used a linear regression model to evaluate the correlation between the obtained% cellular tyrosinase activity and the melanin content of previous experiments. In Figure 6, the authors observed a linear relationship between% intercellular tyrosinase activity and melanin content (R^2^ = 0.98). However, the observed relationship did not include a result from the remedy hexane extract (Figure 6, green spot outside a circle marked as X).

In Figure 6, ten numbers (from 1 to 10) indicated a correlated potency between tyrosinase inhibition and melanin reduction. Using a trend line as a guideline (Figure 6, dot line), one exhibited the most promising candidate against melanin production, while ten showed a negative control (no effect). The two most promising candidates suppressing melanin biosynthesis from the authors’ experiments in this study were highlighted in a gray box, Figure 6 (spot numbers 1 and 2). They were glabridin and the remedy ethyl acetate extract. In both experiments, these candidates exhibited more potency than arbutin, a standard drug (Figure 6, spot number 3). The crude extract remedy and *G. glaba* extract (spot numbers 4 and 6) also showed more substantial activity than the *A. lacucha* extract, a positive reference (spot number 7). However, only the crude extract remedy demonstrated a better melanin suppression effect than kojic acid, another positive control (spot number 5). Water and ethanol extracts of the selected remedy (spot numbers 8 and 9) were the weakest candidates, showing a slight melanin reduction effect above the control (spot number 10).

Even though the statistical model indicated a strong relationship between tyrosinase activity and melanin content, an absolute conclusion could not be made here since there was a significant limitation regarding the experiment. The authors first incubated all samples with cells in the experiment, followed by reading enzymatic activity. Therefore, lowing% intracellular tyrosinase activity might be one of the following situations. First, the samples directly inhibited tyrosinase activity, resulting in melanogenesis suppression. Second, the sample suppressed tyrosinase production via inhibiting signaling pathways, such as blocking the melanocortin-1 receptor (MC1R), and led to a lower melanin content. Finally, the samples exhibited a dual action (directly inhibited tyrosinase activity and suppressed tyrosinase production). Therefore, the authors could not explain the phenomena definitively without further investigation.

### 2.4. Anti-Pigmentation in Zebrafish Larvae

After the promising result in an in vitro enzyme binding and cell-based assays, the authors conducted an in vivo anti-pigmentation in zebrafish larvae of the selected remedy extracts. Unexpectedly, only two remedy extracts (ethanol and water) showed a slight anti-pigmentation effect. All other extracts, including active ingredients (*G. glabra* extract and glabridin), did not suppress melanin production but promoted melanogenesis in zebrafish larvae.

The authors used 1-phenyl-2-thiourea, or PTU, as a positive control, suppressing melanin biosynthesis in zebrafish larvae. As a result (Figure 7), PTU reduced the size of the black spot by nearly half (47% depletion) compared to the 0.03% sea salt solution (negative control).

For the remedy extracts, three extracts, such as crude, hexane, and ethyl acetate extracts, did not shrink the size of the black spot. Instead, they slightly expanded the black spot size from approximately 5% to 17% (Figure 7 and Appendix A). On the other hand, remedy ethanol and water extracts reduced the black spot size from approximately 6% to 15%, showing the best anti-pigmentation effect above all test samples. Notably, the active remedy ingredients, such as *G. glbra* extract and glabridin, enhanced the size of the black spot on zebrafish larvae by up to 35% (Figure 7 and Appendix A). Therefore, this information indicated that glabridin acted as an activator, promoting melanin production in zebrafish larvae. Finally, there was no mathematical relationship between the zebrafish larvae and murine melanoma (B16F1) experiments (Appendix A). Furthermore, the obtained result here was in contrast to the results from the in vitro enzyme binding and cell-based assays earlier.

The findings here were unexpected. All promising bioactive fractions from previous experiments exhibited an insufficient effect against melanogenesis in zebrafish larvae, especially *G, glaba* and glabridin as active ingredients promoting melanin production. However, the authors observed that, in general, the extract from the remedy and *G. glabra* extract as a mixture of metabolites promoted much less melanogenesis effect than glabridin (a pure metabolite). For example, ethyl acetate extract showed a minimum 5% promotion of melanogenesis, while glabridin maximized a 35% promotion effect (Figure 7 and Appendix A). Therefore, the authors hypothesized that another metabolite inside the selected remedy might encounter glabridin-promoting melanogenesis activity. Furthermore, based on the authors’ GC–MS analysis, we suspected that ethyl-*p*-methoxycinnamate might be the glabridin antagonist since it was the major metabolite found in the remedy extracts (Figure 1). Finally, the authors conducted further theoretical experiments to test our hypothesis (following experiments).

### 2.5. In Silico Biology and Molecular Docking Studies

#### 2.5.1. Theoretical Interspecies Glabridin–Tyrosinase Interactions

Due to contradictory results between in vitro (both enzyme-binding and cell-based assays) and in vivo zebrafish larvae studies, the authors conducted a computational experiment seeking a theoretical explanation. The authors first compare the tyrosinase protein structure using multiple techniques, including multiple protein sequences alignment, phylogenetic analysis, and 3D protein structural alignment. Here, the authors aimed to identify conserved and non-conserved regions among tyrosinase used in this study. We used the obtained findings to explain the contradictory results found in the zebrafish assay.

The authors expected that glabridin, as a bioactive metabolite in both *G. glabra* and the selected remedy from in vitro studies, should exhibit a similar effect in an in vivo experiment, inhibiting zebrafish larvae tyrosinase. However, an obtained zebrafish result opposed the biochemical and cell-based assays. Therefore, the authors conducted a bioinformatics study to identify conserved regions among mushroom, murine, and zebrafish tyrosinases used in this study. The authors started by performing a multiple sequences alignment followed by phylogenetic analysis. Eighteen amino acid sequences of tyrosinases and tyrosinase-like proteins were analyzed. In this analysis, tyrosinases and tyrosinase-like proteins from diverse groups of organisms were included, such as fungal, plant, and animal (vertebrate and invertebrate). The phylogenetic analysis indicated that tyrosinase from the three organisms used in this study was highly diverse. Mushroom tyrosinase was located in a different clade (group), away from murine and zebrafish. Murine and zebrafish tyrosinases were highly related. However, murine tyrosinase was closer to human tyrosinase than zebrafish. Finally, the authors provided a phylogenetic analysis of eighteen tyrosinases in a Appendix A. Our finding here was in line with previous reports [18,19].

Later, the authors examined the conserved regions on 3D tyrosinase structures of mushrooms, murine, and zebrafish using the Chimera program (Version 1.11.2). The mavCoservation score was used to determine the degree of conservation among tyrosinases. The obtained result exhibited a high conserved region among three tyrosinases (Figure 8A,B, red color). This conserved region was the active site. On the other hand, most of the structures were moderate to non-conserved regions, as depicted in white and blue colors in Figure 8A,B.

Using a previous report from Chen et al. as a guideline, the authors performed molecular docking between glabridin and mushroom tyrosinase (PDB ID: 2y9x) [20]. Previously, Chen and colleagues reported that glabridin acted as a noncompetitive inhibitor and proposed that glabridin docked at a site near an active site [20]. Therefore, the authors followed Chen’s report [20]. Nevertheless, first, the authors validated our setup docking protocol. As a result, the authors’ docking protocol passed an acceptance criterion (root-mean-square deviation, RMSD, <2), indicating a reliable docking result [21,22]. The validation result is in a Appendix A.

Finally, the authors performed molecular docking on a mushroom, murine, and zebrafish tyrosinase. The best docking pose of each tyrosinase was selected after combining and overlaying docking results of glabridin and its derivatives (3′’,4′’-dihydroglabridin and morusone). Previously, both glabridin derivatives had been reported with an anti-tyrosinase activity. Therefore, overlaying docking results of glabridin and its derivatives was to secure the proper docking pose based on the assumption that similar compounds interacted at the same site. The authors provided the structural alignment of the glabridin derivatives in a Appendix A. Figure 8C,D revealed that the glabridin structure was changed, adapting to each tyrosinase. However, all three models were similar in that glabridin did not interact with an active site of tyrosinase. This finding was in line with Chen’s report, showing glabridin was a noncompetitive inhibitor [20].

In conclusion, the theoretical results indicated that a high diversity among tyrosinase structures across species affected glabridin–tyrosinase interaction, leading to different experimental outcomes found earlier. Therefore, based on an enzyme-inhibitor binding perspective, our findings were able to explain earlier contradictory results between in vitro and in vivo assays.

#### 2.5.2. Proposed Molecular Interplay between Glabridin and Ethyl *p*-methoxycinnamate in Melanin Synthesis via Melanocortin 1 Receptor (MC1R)

Based on the zebrafish experiment earlier, the data indicated that *G. glabra* extract and glabridin activated melanin biosynthesis. However, in the same experiment, the ethanolic extract from the selected remedy suppressed the melanin content slightly lower than the control. This finding hinted that, perhaps, another metabolite (not glabridin) from the remedy extracts suppressed the melanin production enhanced by *G. glabra* and glabridin (Figure 7). Additionally, based on the GC–MS analysis (Figure 1), the authors found a high ethyl *p*-methoxycinnamate in the 80% ethanolic extract. The ethanolic extract exhibited a minimal promoting effect on zebrafish melanogenesis (Figure 8). Therefore, the authors hypothesized that ethyl-*p*-methoxycinnamate might be responsible for suppressing glabridin from promoting melanin production in zebrafish larvae. Thus, the authors conducted another molecular docking experiment to predict a possible molecular interplay between glabridin and ethyl *p*-methoxycinnamate in melanin biosynthesis through the melanocortin-1 receptor (MC1R) for the first time.

An earlier study by Ko et al. in 2014 reported that ethyl *p*-methoxycinnamate inhibited melanin production in a cell-based assay treated with α-MSH [23]. It is commonly known that α-MSH binds at the MC1R, triggering the downstream process for melanin biosynthesis in cells [24]. MC1R is the crucial receptor to trigger the production of melanin. Therefore, the authors compared a binding capacity between glabridin and ethyl *p*-methoxycinnamate on MC1R through a molecular docking experiment.

Until now, the zebrafish MC1R protein structure is not yet available. Therefore, the authors used the predicted zebrafish MC1R protein from the AlphaFold protein structure database [25,26] (accession: Q7ZTA3, accessed on 15 August 2022) as a model. On the other hand, the authors selected a human MC1R protein bound with α-MSH (PDB ID: 7F4D) [27] as a template to visually guide the binding site of α-MSH on the predicted zebrafish MC1R protein (Appendix A). Furthermore, the α-MSH protein structure was extracted from the human MC1R protein crystal structure. Afterward, the authors docked the extracted α-MSH to the selected binding site on the zebrafish predicted MC1R guided by human MC1R protein. This step was to ensure the correct binding pocket selection. Later, the authors used pyDockWEB, an online server [28], to perform protein–protein docking to observe the interaction between α-MSH and the selected binding domain on the zebrafish MC1R obtained from the AlphaFold database earlier. The best docking pose proposed by the online program was selected. The RMSD of the whole zebrafish MC1R plus α-MSH obtained from the docking server was calculated and compared to human MC1R protein bound with α-MSH, our template. The total RMSD value was less than 2 Å (Appendix A), indicating an accepted range to consider as a precise prediction [21,22]. Therefore, the selected binding pocket on the zebrafish MC1R was later used for ligand–protein docking in the following steps.

Next, the authors used Autodock Vina (Version 1.1.2) to evaluate the binding possibility of glabridin (as an activator) and ethyl *p*-methoxycinnamate (as a suppressor) on the zebrafish MC1R [29]. Again, the established docking protocol was validated before performing molecular docking. Next, the authors extracted α-MSH from the original model obtained in the previous paragraph and re-docked it back to the binding pocket. As a result, the established docking parameters could imitate the original model with an RMSD value equal to 1.118 Å, Appendix A [21,22]. Finally, the authors docked two more derivatives of each compound (six compounds in total, Appendix A), helping to identify the best docking pose of each molecule after superpositioning all obtained docking poses [30].

The docking revealed that glabridin bound at the identical pocket as α-MSH, a natural MC1R activator (Figure 9). On the other hand, two possibilities where ethyl *p*-methoxycinnamate could interact with zebrafish MC1R were obtained from the docking experiment. The first possibility was that ethyl *p*-methoxycinnamate shared the same pocket as glabridin and α-MSH. This information would suggest that ethyl *p*-methoxycinnamate competitively competed with glabridin on the zebrafish MC1R receptor, resulting in melanin biosynthesis suppression. On the other hand, the second possibility showed that ethyl *p*-methoxycinnamate docked at the non-active site (allosteric site). Despite the differences, a similar outcome would be expected from both possibilities, suppressing glabridin’s MC1R activation. Even though the authors cannot definitively conclude the mechanism of action, the theoretical simulation via molecular docking indicated that both ethyl *p*-methoxycinnamate and glabridin interacted with the MC1R receptor, either at the active site or the allosteric site or a combination of both. Incorporating the docking result here with the in vivo experiment, the authors proposed the hypothesis that ethyl *p*-methoxycinnamate, as an MC1R antagonist, competed with glabridin, as an MC1R agonist (Figure 9), resulting in compensating melanin content in zebrafish (Figure 8). However, further experiments must proceed to confirm the authors’ hypothesis.

## 3. Discussion

One of the biggest challenges in studying the therapeutic effect of traditional medicine (TM) is determining bioactive metabolite(s), especially in the polyherbal formulation. Even in Thailand, people have been using TM to cure diseases and promote health for centuries [1]. Without knowing bioactive metabolite(s), it is not accessible to understand the therapeutic mechanism of the interested herbal remedy. Therefore, it is difficult to control the quality of a studied remedy leading to a nonrepeatable result and becoming a significant obstacle in TM research. Unlike modern drug studies, a pure compound or metabolite is usually investigated. It is much easier to explore the mechanism of action and control a quality that leads to a more precise outcome. Therefore, many researchers choose to study a pure bioactive metabolite over a polyherbal remedy, causing a knowledge gap. However, in this study, the authors embraced this challenge. We applied a modern scientific approach to investigate the anti-pigmentation activity and determine the bioactive molecules of the selected Thai rejuvenating remedy, No. 11. The authors defined glabridin and ethyl *p*-methoxycinnamate as bioactive molecules inhibiting melanin biosynthesis as presented earlier.

Even though the authors investigated Thai rejuvenating remedy No. 11 for the first time, the anti-pigmentation properties of both bioactive metabolites (glabridin and ethyl *p*-methoxycinnamate) found in the remedy were already reported before by Chen et al. 2016 and Ko et al. 2014 [2,3]. However, it does not mean that the authors’ results lack novelty since both studies investigated the activity of each metabolite as a pure component and studied them separately. On the other hand, in this article, the authors studied both metabolites in a complex fashion, as in polyherbal formulation extracts. Therefore, to the best of the authors’ knowledge, this study is the first report of its kind. Furthermore, there are four extra critical points to be discussed.

First, even though studying a pure bioactive molecule is more advantageous, sometimes a non-desired effect occurs. In this study, the authors’ result demonstrated that glabridin, a pure metabolite derived from *G. glabra* (the active ingredient in the No. 11 remedy), acted as the most potent tyrosinase inhibitor in an in vitro screening. However, when glabridin was tested in an in vivo (zebrafish) model, glabridin distinctively promoted melanin production as a non-desired outcome. Compared to the ethyl acetate extract of remedy No. 11, even though the extract exhibited less anti-tyrosinase activity potency in the screening step, it also came with a minimal non-desired effect in a zebrafish model. Therefore, it is essential to include a non-desired effect when comparing a study of a pure metabolite with an extract, not just considering only a desired one.

Second, the result from this study exhibited that different test models offered different mechanisms of action and were unrelated theoretically. For example, as presented in a previous section, results obtained from mushroom’s and murine’s intracellular tyrosinases activity were in agreement even though they were not related based on phylogenetic (Appendix A) and 3D protein structure (Figure 8) analyses. However, this case did not apply to the results obtained from a murine cell-based assay and a zebrafish experiment, although both tyrosinases were closer related based on phylogenetic analysis (Appendix A). The authors’ findings showed that the zebrafish model might be more suitable for studying an MC1R receptor–metabolite binding rather than a tyrosinase–inhibitor interaction. Therefore, it is vital to consider these differences when comparing the anti-pigmentation effect between test models. 

Third, following the previous paragraph, it is rational to further investigate chemically ethanolic and aqueous remedy No. 11 extracts since both exhibited a visible positive outcome in a zebrafish experiment. It is likely to discover a different group of metabolites derived from the remedy based on different solvent polarities among ethanol, water, and ethyl acetate (a current bioactive fraction). Additionally, it might reveal either a novel bioactive metabolite, a new anti-pigmentation mechanism, or both.

Fourth, in a theoretical molecular simulation experiment, the authors found that ethyl *p*-methoxycinnamate could insert itself into two binding pockets at the MC1R catalytic domain. One was an active site, while another was an allosteric site (non-active site). However, this theoretical experiment was simulated restrictedly at the catalytic domain to save computational resources. Therefore, it might be possible to identify other allosteric binding sites of ethyl *p*-methoxycinnamate on the MC1R receptor if the simulation target domain could be extended. At the same time, following the simulation extension, more computation power will be required. Lastly, it is essential to emphasize that the authors’ simulation here is in line with a previous experimental finding reported by Ko et al. in 2014 [2].

In conclusion, the authors partially filled a knowledge gap by exploring the anti-pigmentation mechanisms of the Thai rejuvenating remedy (a polyherbal formulation) in various experiments, from in vitro to in vivo models. Notably, the authors identified bioactive metabolites inhibiting melanin biosynthesis. Therefore, it can be used in future studies as bioactive markers for a remedy quality control purpose, increasing the result of producibility. Additionally, a computational simulation was integrated. The simulation provided theoretical support for the obtained experimental outcomes and proposed further hypotheses based on scientific findings. Therefore, more investigations are required in the future to confirm the findings and the proposed hypotheses.

## 4. Materials and Methods

### 4.1. Chemical Agents

The authors purchased acetic acid, dimethyl sulfoxide, glacial acetic acid, and HPLC-grade solvents (acetonitrile and methanol) from RCI Labscan Limited, Bangkok, Thailand. While commercial-grade solvents, including ethanol, ethyl acetate, hexane, and methanol, were obtained from Thai Oil Co. Ltd., Bangkok, Thailand. The authors ordered buffers, including disodium hydrogen phosphate and sodium phosphate monobasic dihydrate, from MAY & BAKER Limited Dagenham, London, UK. The authors used standard references such as arbutin, mushroom tyrosinase (Lot #SLBZ0022), sulforhodamine B sodium salt, and trichloroacetic acid (TCA) together with tris(hydroxymethyl)aminomethane from Sigma-Aldrich, Munich, Germany. *Atrocarpus lakoocha* wood (water extract) was self-prepared at the faculty of Pharmaceutical Sciences, Prince of Songkla University, Songkla, Thailand. The authors obtained glabridin from Wuhan ChemFaces Biochemical Co. Ltd., Hubei, China. (3,4-Dimethoxyphenyl)-L-alanine (L-Dopa), hydrochloric acid [HCl] (37% *w*/*w*), and kojic acid were received from Fluka, Sigma-Aldrich, Rockville, USA. Dulbecco’s modified eagle medium (DMEM) and fetal bovine serum (FBS) were purchased from Gibco^™^ Thermo Fisher Scientific, Waltham, USA. D-(+)-glucose anhydrous and Hi-AR/ACS was ordered from HiMedia Laboratories Pvt. Ltd., Mumbai, India. In the meantime, the authors used sodium bicarbonate obtained from Thermo Fisher Scientific India Pvt. Ltd., Hyderabad, India. Potassium chloride and potassium dihydrogen orthophosphate came from Ajax Finechem Pty. Ltd., Sydney, Australia. Later, the authors ordered sodium chloride crystals from J. T. Baker, Bayan Lepas, Malaysia, and 0.25% Trypsin-EDTA (1X) from Gibco^®^ by Life Technologies, Toronto, Canada. Finally, the authors used trypan blue stain (0.4%) from Gibco^™^ Thermo Fisher Scientific, Oxford, UK.

### 4.2. Plant Materials

Sixty-two Thai rejuvenating remedies were selected from commonly known traditional Thai healer textbooks. Dried powder herbal ingredients of all remedies were purchased from the registered traditional Thai herbal pharmacy store in Songkhla province, Thailand. A licensed traditional Thai pharmacist identified the species of all plant materials. A small portion of the remedies and each ingredient were collected as a sample in the authors’ herbarium at the Department of Pharmaceutical Botany and Pharmacognosy, Prince of Songkla University, Songkhla, Thailand.

#### Preparation of Crude Extracts

The dried powder of each Thai rejuvenating remedy was extracted with 80% ethanol using the maceration technique at room temperature for 72 h by following the previous reports [31,32,33]. After that, the solution was filtered with filter paper and completely evaporated with a rotary evaporator (under vacuum at 40 °C) to obtain the crude extracts. The authors repeated the extraction process two more times. Later, the authors combined all three extracts.

The remedy that exhibited the most potency against mushroom tyrosinase was selected for further extraction. Later, the authors applied a series extraction method from non-polar to polar solvents, beginning with hexane, ethyl acetate, ethanol, and water to the selected remedy. The authors used the same extraction procedures as above (room temperature for 72 h). Notably, for water extraction, the protocol was slightly modified. Water extraction proceeded by boiling the samples at 60 °C for 6 h. However, the concentration protocol of the water extract was similar to the other extracts using a rotary evaporator.

### 4.3. Chemical Analysis

#### 4.3.1. Gas Chromatography–Mass Spectrometry Analysis (GC–MS Analysis)

The authors sent the samples to the Office of Scientific Instrument and Testing (OSIT), Prince of Songkla University, for GC–MS analysis. Accordingly, OSIT provided the authors with GC–MS results and an operational condition used in the GC–MS analysis. A 7820A gas chromatograph coupled with a 5975C network mass spectrometer (GC–MS) (Agilent Technologies, Waldbronn, Germany) was the instrument that analyzed all the samples in this study. An Agilent Technologies HP-5MS 5% Phenyl Methyl Silox column (30 m × 250 μm × 0.25 μm) was the separation column used here. The column temperature was initially set at 80 °C and held for 3 min, then increased at a rate of 5 °C/min up to 280 °C and finally held for 5 min. The total run time was 48 min. The injection volume was 1 µL, with a split ratio of 10:1. Helium was used as the carrier gas at a flow rate of 1 mL/min. MS detection was performed with electron ionization (EI) at 70 eV, operating in the full-scan acquisition mode in the m/z range from 35–350. The authors considered only the chemicals presented with more than 1% of the total detected compounds and a percent matching factor of more than 70% from the database.

#### 4.3.2. High-Performance Liquid Chromatography Analysis (HPLC Analysis)

The authors used HPLC analysis to confirm the GC–MS by comparing the detected peak’s retention time (Rt) with a standard reference (Pure compound) from the selected remedy extracts. Shimadzu Prominence-I LC-2030C 3D from Shimadzu Scientific Instruments, INC., Baltimore, U.S.A., was used in this study. The authors modified the HPLC analysis from the previous report [34]. A Hypersil^®^ BDS C18 reversed-phase column (5 µm silica particle size, 250 × 4.6 mm inner diameter) (Thermo Fisher Scientific™, Waltham, MA, USA) was the separation column used. The column temperature was set at ambient temperature. The solvent flow rate was 1.5 mL/min with an injection volume of 20 µL. The elution was carried out with gradient solvent systems, which consisted of water and acetonitrile as mobile phases. Initial gradient elution was from 40 to 45% acetonitrile for between 0 and 30 min. Then, the gradient elution shifted to 40% acetonitrile consistently for between 30 and 35 min. A diode-array UV multi-wavelength detector was used for the detection. Lastly, chromatograms were recorded at 230 nm.

For sample preparation, 10 mg of crude extracts from the selected Thai rejuvenating remedy were dissolved in 1 mL of an appropriate solvent to make a 10 mg/mL concentration. Then, the sample solution was sonicated and filtered through a 0.45 µm syringe filter for HPLC analysis. The authors only adjusted the concentration for standard reference from 10 mg/mL to 0.1 µg/mL, while the other step remained the same as for the crude extract preparation.

### 4.4. Determination of Bioactivities

#### 4.4.1. Determination of Mushroom Tyrosinase Inhibitory Activity (Enzyme-Binding Assay)

For anti-tyrosinase activity, the authors followed an existing protocol previously used by the authors’ laboratory [31,32,33]. The authors briefly describe an anti-tyrosinase experiment here. The authors monitored dopachrome, an intermediate from melanin biosynthesis. Dopachrome absorbed the UV wavelength at 492 nm. Therefore, the authors used a microplate reader to screen and determine anti-tyrosinase activity.

First, the authors added 50 µL of an 80 µg/mL sample solution for the screening step. Then, the authors used kojic acid and water extract from *A. lacucha* wood for positive control at the same concentration. Later, 50 µL of 81.26 unit/mL tyrosinase enzyme solution was added into the wells and incubated at 25 °C for 10 min. For the negative control, the authors used 100 µL of 10 mM phosphate buffer solution (pH 6.8). Finally, 50 µL of 0.85 mM L-Dopa was added, and the optical density (OD) was monitored at 492 nm for 20 min. The authors calculated a percentage of tyrosinase inhibition by using Equation (1):(1)$$\%\;\mathrm{tyrosinase}\;\mathrm{inhibition}=\frac{\left({\mathrm{OD}}_{492}\;\mathrm{of}\;\mathrm{sample}-{\mathrm{OD}}_{492}\;\mathrm{of}\;\mathrm{control}\right)}{{\mathrm{OD}}_{492}\;\mathrm{of}\;\mathrm{control}}\times 100$$

The authors reported the results in a standard form of mean ± SD. We also eliminate the effect from the sample color by deducting the OD_492_ from the well, replacing 50 µL of enzyme solution with 50 µL of 10 mM phosphate buffer solution before applying Equation (1).

For the IC_50_ determination, the authors identified the IC_50_ value by varying each sample’s five concentrations and plotted% tyrosinase inhibition (*Y*-axis) against concentrations (*X*-axis).

#### 4.4.2. Determination of Intracellular Tyrosinase Inhibitory Activity and Melanin Content (Cell-Based Assay)

##### Cell Culture Procedure

Following a previous protocol [32,33], the authors cultivated murine melanoma or B16F1 cells (CLS-400122) (CLS Cell Line Service GmbH, Eppelheim, Germany) in Dulbecco’s Modified Eagle’s medium (DMEM) with 10% heat-inactivated fetal bovine serum, which is called the complete medium solution. First, B16F1 cells were incubated at 37 °C in a humidified atmosphere with 5% CO_2_. After the cells reached approximately 80% confluence, the authors proceeded with further experiments, including cell viability, intracellular tyrosinase inhibitory activity, and melanin content assays.

##### Cell Viability Assay

The authors used an SRB assay for the cell viability assay and followed the previous report [32,33]. First, the cells were seeded in a 96-well plate (8 × 10^3^ cells/well) and incubated at 37 °C in a humidified atmosphere with 5% CO_2_ for 24 h. Later, the cells were treated with a sample solution or 0.1% DMSO as a control. Then, the cells were continually incubated for 48 h. Afterwards, the authors fixed the cells with 100 µL of 10% TCA, kept them at 4 °C for an hour, and were later washed in 10% TCA. After drying, the authors stained the cells with 0.4% SRB solution (0.4% *w*/*v* in 1% acetic acid) and left them at room temperature for 30 min. Finally, the authors added 100 µL of 10 mM Tris base, dissolving SRB color, and used a microplate reader to read the result. The authors read the SRB color at 492 nm. The percentage of cell viability was calculated by Equation (2):(2)$$\%\;\mathrm{cell}\;\mathrm{viability}=\frac{\left({\mathrm{OD}}_{492}\;\mathrm{of}\;\mathrm{sample}\right)}{{\mathrm{OD}}_{492}\;\mathrm{of}\;\mathrm{control}}\times 100$$

##### Intracellular Tyrosinase Assay

The authors determined both intracellular tyrosinase activity and intracellular melanin content by adapted from the previous reports [32,33]. B16F1 cells were cultured in the complete medium solution with 0.25 nM α-MSH. The cells were seeded in a 12-well plate (18 × 10^4^ cells/well) and cultured in a complete medium solution containing 10 nM α-MSH. After the cells were incubated at 37 °C in a humidified atmosphere with 5% CO_2_ for 24 h, the cells were treated with sample and standard (arbutin and kojic acid) solutions. Furthermore, 0.1% DMSO was used as a control. After 48 h of incubation, the treated cells were washed with phosphate buffer saline (PBS) solution and lysed in PBS solution containing 0.1% Triton X-100. Lastly, the cells were centrifuged at 14,000 rpm for 20 min at 4 °C to separate the supernatant solution and cell pellets of lysed cells.

Later, the supernatant was used to determine intercellular tyrosinase activity. First, 100 µL of supernatant solution of lysed cells (treated with 0.1% DMSO, sample, and standard solutions), 100 µL of 2 mg/mL L-Dopa in PBS solution, and 100 µL of 0.1% Triton X-100 in PBS solution were mixed in a 96-well-plate. Afterward, the OD values of the enzymatic reaction in a 96-well plate were immediately measured. Then, a 96-well plate was incubated at 37 °C for an hour. Afterward, the OD values were measured again. As mentioned before, the percentage of intracellular tyrosinase inhibitory activity was calculated with Equation (1).

##### Intracellular Melanin Content

The obtained cell pellets of lysed cells (treated with 0.1% DMSO, sample, and standard solutions), similar to earlier, were dissolved with 1 M NaOH and incubated at 55 °C for an hour. Then, the OD values of the melanin solution were measured at 475 nm. Lastly, melanin concentration was calculated by comparison of the OD values of each sample with a standard curve of standard synthetic melanin.

#### 4.4.3. Pigmentation Inhibitory Activity Assay on Zebrafish Larvae

The authors from the Department of Oriental Medicinal Materials and Processing, College of Life Sciences, Kyung Hee University, South Korea, are researchers who investigated pigmentation inhibitory activity on zebrafish by followed the previous report [32]. Zebrafish (*Danio rerio*) are an animal used to determine the anti-pigmentation effect of the selected remedy. After 9 h post-fertilization (hpf), the zebrafish embryos were placed individually into the 96-well plate containing 100 µL/well of 0.03% sea salt solution and sample solution. In this experiment, the zebrafish embryos were used for 20 embryos per group. Moreover, 10 µg/mL of PTU was used as a positive control group, and 0.03% sea salt solution was used as a control group. After embryos were hatched at 72 hpf, the zebrafish larvae were put on glass slides and embedded using 2% low melting agarose. The dorsal view of the zebrafish larvae was photo-captured to evaluate the black spot size at the zebrafish larvae’s head–dorsal region.

### 4.5. In Silico Biology and Molecular Docking Studies

#### 4.5.1. 2D Multiple Sequences Alignment and Phylogenetic Analysis

All tyrosine amino acid sequences were obtained either from the NCBI database (https://www.ncbi.nlm.nih.gov/, accessed on 22 July 2022) or Uniprot (https://www.uniprot.org/, accessed on 22 July 2022). In addition, accession numbers of all tyrosinase amino acid sequences used in this study are provided in the Appendix A. The Mega-X program, version 10.0.4, was used for multiple alignments (applying the ClustalW package, version 2.1) and phylogenetic analysis (using the minimum evolution method with one thousand pseudoreplicates) [35].

#### 4.5.2. 3D Structural Alignments of Used Tyrosinases in This Study

The authors used a matchmaker package from Chimera program version 1.11.2 to evaluate the residue conservation of three tyrosinases used in this study [36]. All parameters were set as a default value. Additionally, the authors used a mavConservation score to display conserved amino acid residues of three tyrosinase enzymes used in this study. A value of one from a mavConservation score indicates the highest conservation value, while zero indicates a nonconservation value [36,37,38].

#### 4.5.3. Ligand–Protein Docking via Autodock Vina

Nearly all compounds (also called ligands) used in the computational experiment were downloaded from the PubChem database (https://pubchem.ncbi.nlm.nih.gov/, accessed on 1 July 2022) except 3′’,4′’-dihydroglabridin and morusone. The authors used the 3D structure of glabridin obtained from PubChem as a template to generate 3′’,4′’-dihydroglabridin and morusone via the Avocado program [39]. The PubChem ID of all compounds used in this experiment is presented in the Appendix A. The authors followed our previous studies’ protocol to optimize all ligands’ structural geometry and energy using the Avocado program version 1.2.0 [40,41,42]. On the other hand, the protein structure used in this study was obtained either from the Protein Databank or PDB (https://www.rcsb.org/, accessed on 2 July 2022) or the AlphaFold protein structure database for predicted structure (https://alphafold.ebi.ac.uk/, accessed on 2 July 2022) [25,26]. As also mentioned in our reports, the authors used Autodock Tools version 1.5.6 to prepare final docking files for enzyme/receptor and ligands and docking pockets [43]. Before the docking experiment, the authors performed a structural superposition of all protein structures for a better comparison. A crystal structure of mushroom tyrosinase (PDB ID: 2y9x) [44] was used as a reference model for murine (AlphaFold search ID: P11344) and zebrafish tyrosinases (AlphaFold search ID: F1QDZ4) [25,26], while for MC1R, a crystal structure of human MC1R (PDB ID: 7f4d) [27] was used as a template to generate zebrafish MC1R (AlphaFold search ID: Q7ZTA3) [25,26]. As presented below, each enzyme/receptor had a unique binding pocket and docking parameters.

For all tyrosinases, the grid box was set as x = −10.1, y = −28.7, and z = −43.4 with the size of 18 Å × 18 Å × 18 Å.

For the zebrafish MC1R receptor, the grid box was set as x = 69.6, y = 120.3, z = 102.2, with the size of 20 Å × 18 Å × 16 Å.

Default values were applied in nearly all docking parameters, except exhaustiveness adjusted to twenty-four. Before docking, established docking protocols were validated. Finally, twenty docking poses were collected due to each docking experiment.

#### 4.5.4. Vina Post-Docking Analysis

The Viewdock package from the same program used earlier, the Chimera program, was used to visualize the docking results [36]. First, all docking poses obtained from the Autodock vina [29], including the ligands of interest (glabridin and ethyl-*p*-methoxycinnamate) and their derivatives, were overlaid. Later, a conserved docking pose among these ligands was selected and defined as the best pose.

#### 4.5.5. Protein–Protein Docking

pyDockWEB (https://life.bsc.es/pid/pydockweb/, accessed on 7 August 2022) [28] was utilized to identify a binding pocket of α-MSH on the zebrafish MC1R. All docking parameters were set at default. The top ten of the highest docking scores obtained from pyDockWEB were compared to an original binding pocket of an α-MSH on a crystal structure of human MC1R to validate docking results. Only the docking pose with the highest structural similarity to a crystal structure of human MC1R was selected as the best docking pose.

Lastly, the summary of all methods is shown in Figure 1.

### 4.6. Statistic Analysis

Microsoft Excel was used to evaluate a linear relationship between interested parameters and generate a correlation line and R^2^ value [45,46,47].

## Data Availability

Not applicable.

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
