# Peer review of "In Vitro, In Vivo, and In Silico Analyses of Molecular Anti-Pigmentation Mechanisms of Selected Thai Rejuvenating Remedy and Bioactive Metabolites"

_molecules, 2023, doi:10.3390/molecules28030958_

Round 1

Reviewer 1 Report

Dear author(s):

In vitro, in vivo, and in silico analyses of molecular anti-pigmentation mechanisms of selected Thai rejuvenating remedy and bioactive metabolites

After an exhaustive revision, the manuscript is Reconsider after major revision (control missing in some experiments). In general, the study is closely connected to the journal's objectives. The study is very interesting. The English is good. The abstract is good. The introduction is concise and precise, and it has updated references until 2022. The section result is good. The section discussion is an important problem, since is very poor, with a 90% on the description of results and other confusing lines, i.e., the lines correspond to results section, a few on explication of results, a few on comparison with others studies, and nothing on explication (discussion) of the results obtained with respect to other studies. The section materials and methods need a general Figure.

In the following pages, I give a detailed revision of the manuscript.

ABSTRACT

The abstract is good.

1. INTRODUCTION

The introduction is very clear, concise and precise, with good English, and it has updated references until 2022. The authors need to clarify the gap of knowledge, since it is not clear in the introduction.

3. RESULTS

The section of “Results” is characterized by a very detailed description of the results.

The subsection is good, it is very clear, concise and precise, with good English.

3. DISCUSSION

The section of “Discussion” is characterized by an explication of the results, comparison with other studies, and explication (discussion) of the results obtained with respect to other studies.

Lines 481-496. The lines are unnecessary, since the lines correspond to introduction.

Lines 497-511. The lines correspond to results.

Lines 534-550. The lines correspond to results.

Lines 551-559. The lines correspond to conclusions.

In the next lines, the discussions are very difficult to understand, since the lines are hypotheses, nothing concrete.

** The authors must rewrite the entire section. **

4. MATERIALS AND METHODS

General comments

This section is clear. The English is good. The authors must add a Figure that represents all the complete methodology. This Figure will help to understand the methodology.

4.2.1. Preparation of crude extracts

What is the reference?

4.3.1. Gas chromatography-mass spectrometry analysis (GC-MS analysis)

What is the reference?

4.4.2. Determination of intracellular tyrosinase inhibitory activity and melanin content (cell-based assay)

Intracellular tyrosinase assay

What is the reference?

Intracellular melanin content

What is the reference?

4.4.3. Pigmentation inhibitory activity assay on zebrafish larvae

What is the reference?

** The authors need to add a subsection of statistical analysis **

5. CONCLUSIONS

The section should be improved from the suggested changes.

Author Response

Reply to reviewer 1

After an exhaustive revision, the manuscript is Reconsider after major revision (control missing in some experiments). In general, the study is closely connected to the journal's objectives. The study is very interesting. The English is good. The abstract is good. The introduction is concise and precise, and it has updated references until 2022. The section result is good.

The section discussion is an important problem, since is very poor, with a 90% on the description of results and other confusing lines, i.e., the lines correspond to results section, a few on explication of results, a few on comparison with others studies, and nothing on explication (discussion) of the results obtained with respect to other studies.

The section materials and methods need a general Figure.

REPLY: Thank you for your suggestion. Figure of materials and methods in summary was added.

In the following pages, I give a detailed revision of the manuscript.

ABSTRACT

The abstract is good.

REPLY: Thank you.

  1. INTRODUCTION

The introduction is very clear, concise and precise, with good English, and it has updated references until 2022. The authors need to clarify the gap of knowledge, since it is not clear in the introduction.

REPLY: Thank you very much. The gap knowledge was clarified in this part (Lines 43-45).

  1. RESULTS

The section of “Results” is characterized by a very detailed description of the results. The subsection is good, it is very clear, concise and precise, with good English.

REPLY: Thank you very much.

  1. DISCUSSION

The section of “Discussion” is characterized by an explication of the results, comparison with other studies, and explication (discussion) of the results obtained with respect to other studies.

Lines 481-496. The lines are unnecessary, since the lines correspond to introduction.

REPLY: Deleted

Lines 497-511. The lines correspond to results.

REPLY: Deleted

Lines 534-550. The lines correspond to results.

REPLY: Deleted

Lines 551-559. The lines correspond to conclusions.

REPLY: Deleted

In the next lines, the discussions are very difficult to understand, since the lines are hypotheses, nothing concrete.

** The authors must rewrite the entire section. **

REPLY: As the review suggested, the authors followed the suggestion by rewriting the entire discussion section. The authors wrote a new discussion in a way that provided an easier and clearer understanding than the previous one.

  1. MATERIALS AND METHODS

General comments

This section is clear. The English is good. The authors must add a Figure that represents all the complete methodology. This Figure will help to understand the methodology.

REPLY: Thank you for your suggestion. A figure was added as scheme 1 in the materials and methods section.

4.2.1. Preparation of crude extracts

What is the reference?

REPLY: Thank you for your suggestion. The references were added. Plant extraction which was using the maceration technique is very general and well-known. So, it is not necessary to add any references.

4.3.1. Gas chromatography-mass spectrometry analysis (GC-MS analysis)

What is the reference?

REPLY: The authors sent the samples to the Office of Scientific Instrument and Testing (OSIT), Prince of Songkla University, for GC-MS analysis. Then, the OSIT provided the authors with GC-MS results, which were analyzed and compared with GC-MS library. So, it is not necessary to add any references.

4.4.2. Determination of intracellular tyrosinase inhibitory activity and melanin content (cell-based assay) Intracellular tyrosinase assay

What is the reference?

REPLY: Thank you for your suggestion. The references were added.

Intracellular melanin content

What is the reference?

REPLY: Thank you for your suggestion. The references were added.

4.4.3. Pigmentation inhibitory activity assay on zebrafish larvae

What is the reference?

REPLY: Thank you for your suggestion. The reference was added.

** The authors need to add a subsection of statistical analysis **
REPLY: As the reviewer suggested, the authors added a subsection for our statistical analysis.

  1. CONCLUSIONS

The section should be improved from the suggested changes.

REPLY: After rewriting a new discussion, the author reduced the size of the discussion by half compared to the previous version and decided to integrate a conclusion into the new discussion making it more precise. Additionally, based on the molecule format, the conclusion in optional. Therefore, the conclusion

Reviewer 2 Report

The authors performed a series of analysis of selected Thai rejuvenating remedy and aiming to explore the mechanism. 

Overall, the design and hypothesis are sounding, and the authors has put a lot of efforts trying to elucidate the mechanism of the Thai traditional medicine.

However, during the reading, I feel some of the contents were not organized very well for understanding.  I'll point out my concerns as follows and hope the authors could address them correspondingly.

1.  Line 52-54. "It causes pigment overproduction.... it can lead to other severe medical conditions like skin cancer." It's pretty confusing whether "it" here indicates to "solar lentigo", or its risk factors "aging and UV exposure".

2. Table 1's title need to be more specific and different from table 2. 

3. Section 2.2 needs to be condensed. Pie charts seems to be unnecessary. I don't see the point of explicitly reporting the HPLC result. 

4. Line 167, replace "such as" with "including". It's not 3 random examples, it's the only 3 positive results.

5. Section 2.3.4, I don't understand the rationale behind the correlation analysis. Tyrosinase is already known to be necessary for the production of melanin. There is definitely correlation relationship between these 2 variables, no need to prove it. 

6. I'd like to see more discussion about the 'unexpected results" in zebrafish larvae.

7. Section 2.5.1, line 363-375. No point of using a whole paragraph to report the repeating of an already published docking job. Also, each docking job uses different random seeds to perform the analysis, so you'll never get exactly same results when repeating it, but a same trend of interactions. 

8. For the model generated by AlphaFold, I wonder if any geometry regulation was done before using it for docking. 

Author Response

Reply to reviewer 2

The authors performed a series of analysis of selected Thai rejuvenating remedy and aiming to explore the mechanism. Overall, the design and hypothesis are sounding, and the authors has put a lot of efforts trying to elucidate the mechanism of the Thai traditional medicine.

However, during the reading, I feel some of the contents were not organized very well for understanding.  I'll point out my concerns as follows and hope the authors could address them correspondingly.

  1. Line 52-54. "It causes pigment overproduction.... it can lead to other severe medical conditions like skin cancer." It's pretty confusing whether "it" here indicates to "solar lentigo", or its risk factors "aging and UV exposure".

REPLY: All pronouns were revised. In the first sentence, "It" indicates "the factors of aging and UV exposure." Therefore, the authors revised "it" into "these factors" for a better understanding.

  1. Table 1's title need to be more specific and different from table 2.

REPLY: The authors renamed the title of Table 1, which is now more specific and distingue from Table 2's title.

  1. Section 2.2 needs to be condensed. Pie charts seems to be unnecessary. I don't see the point of explicitly reporting the HPLC result.

REPLY: Thank you for your suggestion, the authors would like to show the pie charts in order to  to conclude all chemical components in selected remedy's extracts including the major and minor metabolites of this remedy by GC-MS analysis.

Since, glabridin is the active ingredient in the selected remedy's extracts. Then, the point to reporting the HPLC result is for confirmation of the presence of glabridin in this selected remedy's extracts. So, the section 2.2 is no need to be condensed.

  1. Line 167, replace "such as" with "including". It's not 3 random examples, it's the only 3 positive results.

REPLY: Thank you, it was already revised.

  1. Section 2.3.4, I don't understand the rationale behind the correlation analysis. Tyrosinase is already known to be necessary for the production of melanin. There is definitely correlation relationship between these 2 variables, no need to prove it.

REPLY: The reviewer correctly mentioned that tyrosinase activity is correlated to melanin production. Therefore, the authors provided three reasons why the authors evaluated the correlation addressing the reviewer's question.

First, as presented in the manuscript, not all authors' test samples showed the inhibition of melanin production by reducing tyrosinase activity like hexane extract (presented as X in the figure). Instead, it hinted that there might be another mechanism in down-regulating melanin biosynthesis. Therefore, it does not always mean lowering melanin content should reduce tyrosinase activity. However, the authors did not point this out because it was not within our scope.

Second, even though obtained remaining tyrosinase activity was correlated statistically with melanin content from an experiment, it was not directly implied that the samples inhibited enzyme activity. Instead, the samples might have disrupted tyrosinase production by inhibiting relevant signaling pathways. Therefore, the correlation helped the authors to emphasize this point.

Third, the correlation line also helped the reader to visualize the potency of all test samples quickly and comprehensively in both experiments.

These three reasons are the reasons why the authors added the correlation in the manuscript.

  1. I'd like to see more discussion about the 'unexpected results" in zebrafish larvae.

REPLY: The authors rewrote the entire discussion section as another reviewer requested to be more concise and precise. Therefore, the authors minimize the content. However, the author proposed that zebrafish larvae may be more suitable for the MC1R study than investigating tyrosinase activity. To inform the reviewer, the authors plan to investigate these unexpected results further for more detail in our future study. However, the authors decided not to discuss the unexpected results further in this current manuscript.

  1. Section 2.5.1, line 363-375. No point of using a whole paragraph to report the repeating of an already published docking job. Also, each docking job uses different random seeds to perform the analysis, so you'll never get exactly same results when repeating it, but a same trend of interactions. 

REPLY: The authors deleted this paragraph as suggested by the reviewer.

  1. For the model generated by AlphaFold, I wonder if any geometry regulation was done before using it for docking. 

REPLY: The authors did not add an additional geometry regulation further than the default set by the docking program. Also, the authors did not perform any further geometric modification as mentioned in the manuscript, from the protein structure obtained from the AlphaFold. Since the authors found that the 3D structure of the AlphaFold zebrafish MC1R protein was close to the crystal structure of human MC1R protein, as mentioned in the manuscript, after comparing a 3D structural alignment (Figure S9).

Reviewer 3 Report

In this manuscript the authors decided to screen the anti-mushroom tyrosinase activity of sixty-two Thai rejuvenating remedies since they have been claimed to have anti-aging and restoring skin effects. They evaluated the anti-pigmentation effect at the molecular level of the selected Thai rejuvenating remedy to fulfill the knowledge gap. They selected the most promising remedy to analyze further chemically and biologically. They found that the selected remedy showed promising activity against the enzyme tyrosinase with an IC50 value 2.4 times higher than that of kojic acid (positive control). They identified glabridin as a bioactive molecule against tyrosinase with an IC50 value lower than that of kojic acid. They reported a molecular anti-pigmentation mechanism of the selected Thai rejuvenating remedy by combining the results from in silico, in vitro, and in vivo experiments. Based on the gas chromatography coupled with mass spectroscopy (GC-MS) analyses, they found that the ethyl-p-methoxycinnamate is the primary metabolite in remedy extracts. The authors proposed that ethyl-p-methoxycinnamate suppresses glabridin and promotes melanin production effect. They will investigate further by observing an expression of the MC1R downstream signaling pathway of zebrafish treated with glabridin and ethyl-p-methoxycinnamate.

I have some questions and suggestions:

1) Chemical structures and IUPAC names of kojic acid, glabridin and ethyl-p-methoxycinnamate were not included.
2) Water is considered as an important solvent as its molecules serve as the interactive bridges between the protein and its inhibitor (ligand). Therefore, what is the importance of water molecules in the binding free energy calculations?.
3)There is not Supporting Information.

The work is interesting, it contains studies in silico, in vitro and in vivo experiments. I think it is suitable for the journal.

Author Response

Reply to reviewer 3

In this manuscript the authors decided to screen the anti-mushroom tyrosinase activity of sixty-two Thai rejuvenating remedies since they have been claimed to have anti-aging and restoring skin effects. They evaluated the anti-pigmentation effect at the molecular level of the selected Thai rejuvenating remedy to fulfill the knowledge gap. They selected the most promising remedy to analyze further chemically and biologically. They found that the selected remedy showed promising activity against the enzyme tyrosinase with an IC50 value 2.4 times higher than that of kojic acid (positive control). They identified glabridin as a bioactive molecule against tyrosinase with an IC50 value lower than that of kojic acid. They reported a molecular anti-pigmentation mechanism of the selected Thai rejuvenating remedy by combining the results from in silico, in vitro, and in vivo experiments. Based on the gas chromatography coupled with mass spectroscopy (GC-MS) analyses, they found that the ethyl-p-methoxycinnamate is the primary metabolite in remedy extracts. The authors proposed that ethyl-p-methoxycinnamate suppresses glabridin and promotes melanin production effect. They will investigate further by observing an expression of the MC1R downstream signaling pathway of zebrafish treated with glabridin and ethyl-p-methoxycinnamate.

I have some questions and suggestions:

1) Chemical structures and IUPAC names of kojic acid, glabridin and ethyl-p-methoxycinnamate were not included.

REPLY: The chemical structures of all compounds were presented in Table 2, and IUPAC names of Kojic acid (line 102), glabridin (line 107), and ethyl p-methoxycinnamate (line 131) were added as the reviewer's suggested.

2) Water is considered as an important solvent as its molecules serve as the interactive bridges between the protein and its inhibitor (ligand). Therefore, what is the importance of water molecules in the binding free energy calculations?

REPLY: As the reviewer mentioned, the waters play a vital role in interactions between amino acid residues in the active site and ligand. However, in classical docking, as the authors used in this study, the water molecules are ignored to reduce computational resources. Recently, research articles, for example, Xiao et al. (2018), reported that allowing water molecules in the active site during docking improved docking accuracy. However, it was not always distinguishable. In some cases, there were only slight differences between removing and allowing water molecules in the docking site. The only difference was that allowing water molecules caused more computational resources. Additionally, in the protocol published in Nature protocol in 2021, the authors of the Nature protocol suggested that the water molecules in the binding site can remain there if the system allows it. However, this is not obligated.

             Finally, we, the authors of this manuscript, validated our established docking protocol by redocking the native ligand back to its original position after extracting it from the protein crystal structure (water molecules were presented in the crystal structure). In our docking, the waters were removed from the active site. After re-docking, the authors compared the re-docking result of the native ligand (without water molecules) with its original position on the crystal structure (with water molecules). The authors found that our docking provided a generally acceptable accuracy with RMSD in less than 2 Å. Therefore, the effect of water molecules in our docking setup is minimal.

             To answer the reviewer's question, the authors' docking did not take water interactive bridges in a binding affinity calculation because Authodock vina did not allow it. Also, as the reviewer may notice in the manuscript, the authors did not report the binding affinity from the docking result because our aim here was to identify potential binding sites first. Later, in future work, the author will consider binding affinity to evaluate which binding site is preferable. Therefore, the docking binding affinity did not include in the current manuscript.

References:

Xiao, W., Wang, D., Shen, Z., Li, S., & Li, H. (2018). Multi-body interactions in molecular docking program devised with key water molecules in protein binding sites. Molecules23(9), 2321.

Bender, B. J., Gahbauer, S., Luttens, A., Lyu, J., Webb, C. M., Stein, R. M., ... & Shoichet, B. K. (2021). A practical guide to large-scale docking. Nature protocols16(10), 4799-4832.

3)There is not Supporting Information.

REPLY: The authors would like to apologize for forgetting to upload the supplementary file. The author is now working with the corresponding author to upload our supplementary file. Please contact the editorial team if the reviewer can not access our supplementary file.

The work is interesting, it contains studies in silico, in vitro and in vivo experiments. I think it is suitable for the journal.   

REPLY: The authors would like to thank the reviewer for the reviewer's compliment and recommendation.

Round 2

Reviewer 1 Report

Dear Author(s)

After an exhaustive revision, the manuscript is Accept in present form. The resubmitted manuscript has been completely improved compared to its previous version. Therefore, the manuscript can be published in “Molecules”.

Best regards
